# Identification of Southeast Asian *Anopheles* mosquito species using MALDI-TOF mass spectrometry

Victor Chaumeau[1,2]°*, Martine Piarroux[3]°, Thithiworada Kulabkeeree[1], Sunisa Sawasdichai[1], Aritsara Inta[1], Wanitda Watthanaworawit[1], François Nosten[1,2], Renaud Piarroux[3], Cécile Nabet[3]°

1 Faculty of Tropical Medicine, Shoklo Malaria Research Unit, Mahidol-Oxford Research Unit, Mahidol University, Mae Ramat, Thailand, 2 Nuffield Department of Medicine, Centre for Tropical Medicine and Global Health, University of Oxford, Oxford, United Kingdom, 3 Institut Pierre-Louis d'Epidémiologie et de Santé Publique, Inserm, IPLESP, AP-HP, Groupe Hospitalier Pitié-Salpêtrière, Service de Parasitologie-Mycologie, Sorbonne Université, Paris, France

° These authors contributed equally to this work.
* victor@shoklo-unit.com

**Data Availability Statement:** All relevant data are within the manuscript and its Supporting Information files.

## Abstract

Malaria elimination in Southeast Asia remains a challenge, underscoring the importance of accurately identifying malaria mosquitoes to understand transmission dynamics and improve vector control. Traditional methods such as morphological identification require extensive training and cannot distinguish between sibling species, while molecular approaches are costly for extensive screening. Matrix-assisted laser desorption and ionization time-of-flight mass spectrometry (MALDI-TOF MS) has emerged as a rapid and cost-effective tool for *Anopheles* species identification, yet its current use is limited to few specialized laboratories. This study aimed to develop and validate an online reference database for MALDI-TOF MS identification of Southeast Asian *Anopheles* species. The database, constructed using the in-house data analysis pipeline MSI2 (Sorbonne University), comprised 2046 head mass spectra from 209 specimens collected at the Thailand-Myanmar border. Molecular identification via COI and ITS2 DNA barcodes enabled the identification of 20 sensu stricto species and 5 sibling species complexes. The high quality of the mass spectra was demonstrated by a MSI2 median score (min-max) of 61.62 (15.94–77.55) for correct answers, using the best result of four technical replicates of a test panel. Applying an identification threshold of 45, 93.9% (201/214) of the specimens were identified, with 98.5% (198/201) consistency with the molecular taxonomic assignment. In conclusion, MALDI-TOF MS holds promise for malaria mosquito identification and can be scaled up for entomological surveillance in Southeast Asia. The free online sharing of our database on the MSI2 platform (https://msi.happy-dev.fr/) represents an important step towards the broader use of MALDI-TOF MS in malaria vector surveillance.

**Funding:** "This research was funded by Wellcome (#220211), the Bill and Melinda Gates Foundation (#OPP1177406) and the Global Fund (#QSE-M-UNOPS-20864-003-32). There was no additional external funding received for this study".

**Competing interests:** The authors have declared that no competing interests exist.

## Introduction

The increasing prevalence of mosquito-borne diseases worldwide underscores the need to strengthen vector control capabilities [1]. Malaria control remains a significant challenge in the World Health Organization South-East Asia Region, representing 33% of the global burden outside Africa [2]. Despite notable progress, with a decrease in cases, 5.2 million cases were reported in 2022, with *P. vivax* responsible for 51% of them. Political instability in Myanmar has resulted in increases in both *P. falciparum* and *P. vivax* cases, significantly impacting malaria control efforts in Thailand. In 2022, Thailand has experienced a 158% rise in reported cases compared to 2021, underscoring the urgent need for enhanced malaria vector surveillance along the Thailand-Myanmar border. The surge in malaria observed in recent years on the Thailand-Myanmar border is multi-factorial and remains largely unexplained. While entomological factors may play a role, current data are lacking. Political instability in Myanmar has impaired health services and affected human behaviors. Specifically, armed-conflict has disrupted access to early diagnosis and treatment, crucial for falciparum malaria elimination [3], and likely increased human exposures to vector bites. Indeed, fleeing civilians sought temporary shelters in forested areas and on the river banks which delimit the international border between Thailand and Myanmar, two typical habitats of the main local vectors [4]. Mosquito bed nets have only a marginal impact on malaria in this region [5–7] because the vectors bite mostly outdoors and at a time when people are not protected by mosquito bed nets [8–10]. Parasitological factors including a shift in *Plasmodium* spp. resistance to antimalarial drugs may also be involved [11,12] and are the focus of active surveillance.

The distinctive ecological characteristics of this border region contribute to one of the highest diversity of malaria mosquito species in the world [13]. Endemic species are distributed among 18 major groups within the subgenera *Anopheles*, *Baimaia*, and *Cellia* [14]. Accurate identification of mosquito vectors is critical for understanding transmission dynamics and assessing the effectiveness of vector control interventions, especially in the current context of global changes that may exacerbate the burden of mosquito-borne diseases [15,16]. However, this task is challenging due to the morphological indistinguishability of many closely related species, which form complexes of cryptic or sibling species [17]. Additionally, the taxonomy of some of these complexes remains unresolved due to the difficulty in determining species status.

For large-scale entomological investigations, there is a need for accurate, affordable, and rapid identification tools. Molecular approaches, including restriction fragment length polymorphism, allele-specific PCR, and Sanger sequencing, have been proposed to identify sibling mosquito species [18–20]. Despite their effectiveness, these methods can be slow and costly to implement. To address these challenges, matrix-assisted laser desorption/ionization time-of-flight mass spectrometry (MALDI-TOF MS) has emerged as an innovative tool for arthropod vector studies [21]. Known for its cost-effectiveness and simplicity this protein fingerprinting-based method has already revolutionized clinical microbiology by facilitating rapid identification of microorganisms [22]. In the case of *Anopheles* mosquito species, identification is achieved by comparing the mass spectra of a protein extract from a dissected anatomical part to a reference database. Combined with machine learning, it holds promise for the identification of sibling species [23–25] and the assessment of entomological factors influencing malaria transmission, such as *Plasmodium* infection rates, past blood meal, and mosquito age [26]. MALDI-TOF MS reference databases have been established for the identification of *Anopheles* species from different parts of the world [27–31], including from Southeast Asia [32,33]. However, these databases are often limited in species coverage and the access is usually restricted to the authors of the respective studies.

In this study, we aimed to establish a MALDI-TOF MS reference database for the accurate and rapid identification of *Anopheles* mosquito species endemic to the Thailand-Myanmar

border. Notably, we made this database accessible online through the MSI2 platform, a free web application developed at the Sorbonne University for the MALDI-TOF MS identification of medically important fungi, parasites, and arthropods [34–36]. The sharing of reference databases through such accessible online platforms will facilitate the wider adoption and advancement of entomological research using MALDI-TOF MS, as for molecular databases.

## Materials and methods

### Sample collection

Entomological surveys were conducted across 16 villages in the Karen (Kayin) state of Myanmar between November 16, 2020, and May 7, 2021 (Fig 1). Mosquitoes were captured in 5-mL plastic tubes using the animal-baited trap (buffalo, cow, or goat) collection method and transported to the Shoklo Malaria Research Unit at the end of the survey period. Upon arrival at the laboratory (1 to 7 days after collection), mosquitoes were macroscopically classified at the genus level, and *Anopheles* were morphologically identified at the group level using a dichotomic identification key [37]. A subset of these malaria mosquitoes was randomly selected to create a panel representative of the different villages and species diversity. Specimens were

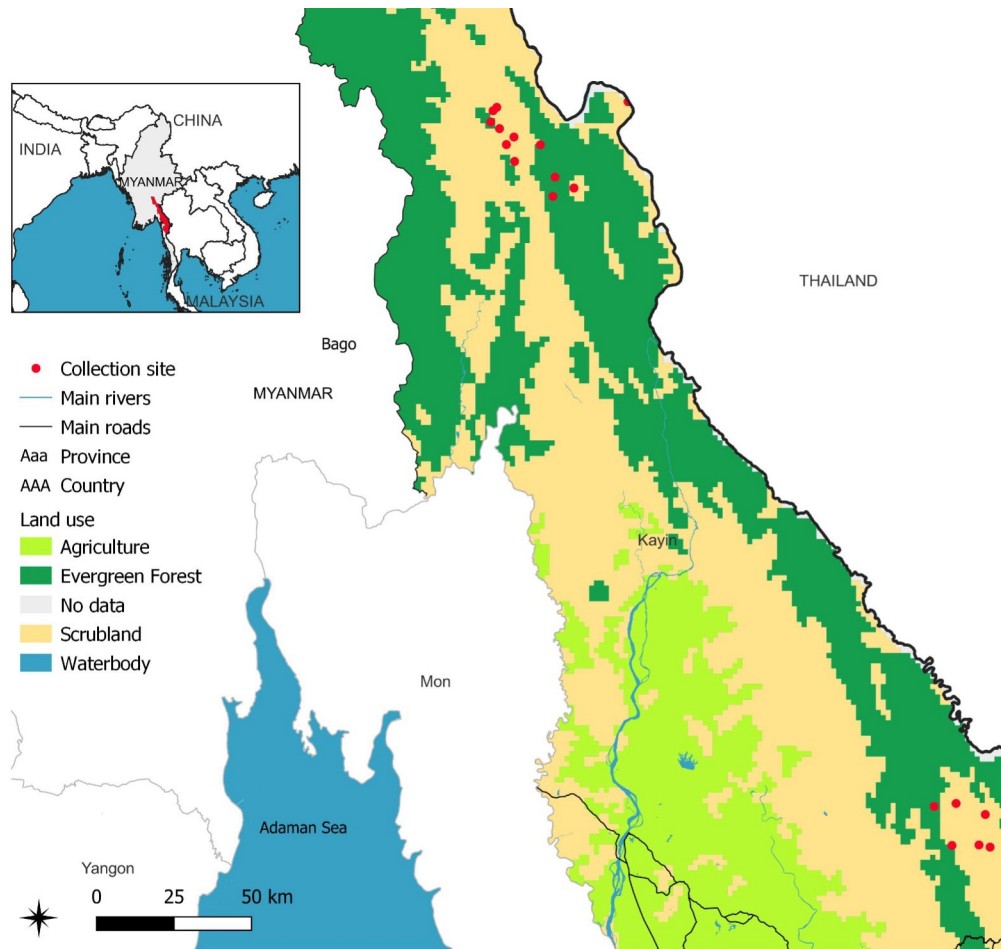

**Fig 1. Location of *Anopheles* collection sites for MALDI-TOF MS analysis in the Kayin province, Thailand-Myanmar boarder.** Base map and data from OpenStreetMap and OpenStreetMap Foundation.

dissected on a glass slide with a set of minutien pins. Head and abdomen were taken separately in 1.5-mL plastic tubes and stored at -80˚C until further processing.

## DNA extraction and PCR amplification

DNA was extracted from dissected mosquito abdomens using the cetyl trimethylammonium bromide method as previously described [38]. Amplification of cytochrome c oxidase subunit I (COI) was performed using the primer pair LCO1490 (5'-GGT CAA CAA ATC ATA AAG ATA TTG G-3') and MTRN (5'-AAA AAT TTT AAT TCC AGT TGG AAC AGC-3') [39,40]. The PCR mix consisted of 1X Goldstar™ DNA polymerase (Eurogentec, Seraing, Belgium) and 400 nM of each primer. PCR was performed in a total reaction volume of 25 μl (4 μl of DNA template diluted at 1: 100 in PCR grade water and 21 μl of PCR mix). The thermocycling protocol consisted of an initial activation step of 1 min at 94˚C, followed by 40 amplification cycles of 20 s at 94˚C, 20 s at 51˚C and 30 s at 72˚C. Reactions that failed to amplify the target were repeated with the reverse primer HCO2198 (5'-TAA ACT TCA GGG TGA CCA AAA AAT CA-3') [40] using the same reaction conditions. Amplification of internal transcribed spacer 2 (ITS2) was performed with the primer pair ITS2A (5'-TGT GAA CTG CAG GAC ACA T-3') and ITS2B (5'-ATG CTT AAA TTY AGG GGG T-3') [41] and using the same reaction conditions, except for the primer concentration (100 nM each). The PCR products were purified using the illustra™ ExoProStar 1-Step commercial kit following the manufacturer's instructions. Sanger sequencing of the purified product was outsourced to Macrogen™ (Seoul, South Korea) and performed using the forward primer (Genbank accesion numbers: PP339876 –PP340064 and PP372871—PP373055). If sequencing of both COI and ITS2 failed, the sample was excluded.

## Protein extraction and mass spectra acquisition

Proteins were extracted from dissected heads as described previously [34] with some modifications. Dissected heads were put into 1.5-mL microcentrifuge tubes and rinsed in 70% ethanol for 10 min. The tubes were centrifuged at 18,000 g for 10 min and the supernatant was discarded. After a second centrifugation at 18,000 g for 2 min, the remaining ethanol solution was removed using a micropipette and left to evaporate until dry. Protein extraction was performed by adding 10 μL of 70% formic acid solution (bioMérieux, Lyon, France, catalog number: 411072). After manual homogenization with a micropipette, the homogenates were incubated for 5 min at room temperature. Then, 10 μL of 100% acetonitrile (VWR, Randor, USA, catalog number: 20060.32) were added and the samples were incubated for an additional 5 min. The homogenates were then centrifuged at 18,000 g for 2 min, and 1 μL of the protein extracts was deposited onto a disposable target plate (bioMérieux, catalog number: 410893). Once dried, the deposits were covered with 1 μL of alpha-cyano4-hydroxycinnamic acid matrix (bioMérieux, catalog number: 411071). Ten spectra were made for each specimen. Mass spectra were acquired with a Vitek MS (bioMérieux, Lyon, France) in the RUO mode using the Shimadzu Biotech Launchpad MALDI-TOF MS application (Shimadzu Biotech, Kyoto, Japan). The spectra were acquired in linear mode in ion-positive mode at a laser frequency of 60 Hz and a mass range of 2–20 kDa. *Escherichia coli* ATCC 8739 was used as a control calibration for each run following manufacturer's instruction. Raw data files were exported in mzXML format and these files were used for subsequent analysis.

## *Anopheles* species identification with DNA sequence data

The Sanger chromatograms were manually trimmed and inspected using Unipro UGENE software version 48 [42]. This process aimed to retain only the clean portions of each sequence

and correct artifactual polymorphisms. In the phylogenetic analysis of ITS2 sequences, information about secondary structure was implemented by annotating ITS2 using the ITS2 annotator (a 5.8S-28S rRNA interaction and an HMM-based annotation program available online) [43]. To identify mosquito species, COI and ITS2 sequences were queried against the NCBI Nucleotide Collection (nr / nt) database using BLASTn [44]. A match to a mosquito species was established when the identity between the query and subject sequences reached 98%. Additionally, COI sequences were queried against the Barcode of Life Data System (BOLD), which encompasses a larger collection of COI sequences compared to the NCBI Nucleotide database [45].

To further validate BLAST and BOLD identification results, a phylogenetic analysis was conducted using the study sequences and reference sequences sourced from Genbank. COI sequences were aligned with Clustal W version 2.1 [46] and ITS2 sequences were aligned with MAFFT using the X-INS-i algorithm and default parameters [47]. The phylogenetic analysis was performed in MrBayes v3.2, using a general time-reversible substitution model and gamma rates [48]. In the analysis of COI sequences, the dataset was partitioned to estimate different mutation rates for the two first and the third codon positions. Each analysis comprised two independent runs with four chains, running for 1,000,000 generations with a sample frequency of 100 generations. The first 25% trees were discarded as burn-in, and posterior probabilities were estimated from the remaining trees to infer branch support.

When DNA sequence analysis failed to discriminate between sibling species, the sample was labeled with the name of its species pair or complex (ITS2: *An. annularis* s.l., *An. campestris/wejchoochotei*, *An. culicifacies* s.l. and *An. tessellatus* s.l.; COI: *An. annularis* s.l., *An. baimaii/dirus*, *An. campestris/wejchoochotei*, *An. culicifacies* s.l., *An. kleini/sinensis* and *An. tessellatus* s.l.). Results from COI and ITS2 sequence analyses were combined to label samples at the species level when possible (e.g., specimens identified as *An. baimaii* with ITS2 and as *An. baimaii/dirus* with COI were labeled as *An. baimaii*).

## Construction of MSI2 reference database

Spectra were visualized and preprocessed using standard procedure including smoothing (Python library pimzml), baseline correction, and custom peak picking with a selection of the 70 highest intensity peaks. A similarity score was calculated for each pairwise spectra comparison using the MSI2 application [49]. Using this algorithm, the range of possible score values is comprised between 0 (indicating no similarity) and 100 (indicating complete similarity). The corresponding identification result corresponds to the reference spectrum that yields the highest score. Each reference mass spectrum was taxonomically identified using the stringent molecular identification criteria described in the previous section.

We then developed a decision algorithm to select high-quality spectra. To account for intra-specimen reproducibility during the construction of the MSI2 reference database, pairwise comparisons were conducted between technical replicates of the same sample. Spectra with at least one intra-specimen score value <40 were excluded from the MSI2 reference database. For assessing inter-specimen reproducibility, pairwise comparisons were carried out between all spectra of the same mosquito species. Except for rare species with fewer than four specimens, spectra with at least one inter-specimen score value <30 were excluded from the MSI2 reference database. Finally, if more than five technical replicates of a sample were affected by the preceding criteria, all spectra of that sample were excluded from the MSI2 reference database.

To visualize the reproducibility of the spectra in the final MSI2 reference database, a heat map grid of the score values was constructed between each spectrum obtained using MSI2.

This was based on the mean value of the similarity scores obtained from the different spectrum replicates of each specimen.

## Evaluation of the performance of MALDI-TOF MS identification

The performance of the MSI2 reference database for the identification of *Anopheles* species was evaluated using a test panel. This panel consisted of four technical replicates per specimen, selected from the initial *Anopheles* dataset in sequential order of spotting. The test panel was then compared to the MSI2 reference database, excluding pairwise comparisons between technical replicates of the same specimen. The MSI2 identification of a tested specimen was determined by identifying the specimen in the MSI2 reference database that obtained the best score among the four replicates tested, as previously published [35].

Subsequently, the MALDI-TOF MS identifications were compared to the molecular identifications for each tested specimen. Since pairwise comparisons between technical replicates of the same specimen were discarded, species represented by a single specimen in the MSI2 reference database served as negative controls, mimicking queries of unreferenced species.

To evaluate the performance of the MALDI-TOF MS identification system, different score thresholds were applied, above which results were considered interpretable. The performance metrics included the proportion of identified spectra, *i.e.*, spectra with a score value above the threshold, and the positive predictive value (PPV), *i.e.*, the probability that a MALDI-TOF MS identification result is accurate.

## Ethics

The study was approved by the Oxford Tropical Research Ethics Committee, the Karen Department of Health and Welfare, Karen National Union and the Tak Province Border Community Ethics Advisory Board [50]. The land accessed is protected by the local Karen authorities, and no sampling of sensitive animals or plants occurred.

## Results

### Molecular identification and MALDI-TOF MS taxonomic assignment

Of the 228 *Anopheles* specimens selected for this study, 214 could be molecularly identified based on the analysis of ITS2 (29 specimens), COI (25 specimens), or both markers (160 specimens); 14 specimens were excluded because the sequencing of ITS2 and COI failed. Based on these DNA sequence data, specimens in the dataset were assigned to 20 *sensu stricto* species and 5 sibling species pairs or complexes (Tables 1 and S1).

One specimen of the Annularis Group had a COI sequence with <98% similarity to the sequences of the NCBI Nucleotide and BOLD databases, but an ITS2 sequence identical to that of *An. annularis* s.l. in the NCBI Nucleotide database. Therefore, it was labeled as a putatively new species in this species complex (*An.* sp. near *annularis*). Two main lineages of *An. minimus* were identified based on the analysis of COI sequences and labeled as *An. minimus* Clade I and *An. minimus* Clade II.

For a specific complex, when the MSI2 reference database contained only one species, the MALDI-TOF MS identification result could not be specified at the species level. Instead, it was assigned at the complex level for all identifications within that particular complex. For instance, since there were no *An. harrisoni* spectra in the MSI2 reference database but only *An. minimus* s.s., if the MALDI-TOF MS identification result was a reference specimen molecularly identified as *An. minimus* s.s., MSI2 would answer *An. minimus* s.l. The phylogenetic trees are provided in S1 and S2 Figs.

**Table 1. Molecular identification of Southeast Asian *Anopheles* species and taxonomic assignment by MALDI-TOF MS.**

| Subgenus | Group | Species (ITS2 and or COI molecular identification) | Level of MALDI-TOF MS taxonomic assignment | No. villages collected | No. specimens in MSI2 database (n = 209) | No. specimens in the test panel (n = 214) |
|---|---|---|---|---|---|---|
| Anopheles | Barbirostris | *An. campestris/wejchoochotei* | *An. campestris/wejchoochotei* | 1 | 1 | 1 |
| Anopheles | Barbirostris | *An. dissidens* | *An. dissidens* | 12 | 19 | 21 |
| Anopheles | Barbirostris | *An. saeungae* | *An. saeungae* | 2 | 4 | 4 |
| Anopheles | Hyrcanus | *An. sinensis* | *An. kleini/sinensis* | 9 | 12 | 12 |
| Anopheles | Hyrcanus | *An. peditaeniatus* | *An. peditaeniatus* | 3 | 3 | 3 |
| Cellia | Annularis | *An. annularis s.l.* | *An. annularis s.l.* | 5 | 9 | 9 |
| Cellia | Annularis | *An. sp. nr. annularis* | *An. annularis s.l.* | 1 | 1 | 1 |
| Cellia | Annularis | *An. nivipes* | *An. nivipes* | 8 | 11 | 11 |
| Cellia | Annularis | *An. philippinensis* | *An. philippinensis* | 4 | 4 | 4 |
| Cellia | Funestus | *An. culicifacies s.l.* | *An. culicifacies s.l.* | 7 | 12 | 13 |
| Cellia | Funestus | *An. jeyporiensis* | *An. jeyporiensis* | 3 | 3 | 3 |
| Cellia | Funestus | *An. minimus* | *An. minimus s.l.* | 3 | 4 | 4 |
| Cellia | Funestus | *An. minimus* clade I | *An. minimus s.l.* | 5 | 6 | 6 |
| Cellia | Funestus | *An. minimus* clade II | *An. minimus s.l.* | 16 | 25 | 25 |
| Cellia | Funestus | *An. varuna* | *An. varuna* | 1 | 2 | 2 |
| Cellia | Jamesii | *An. jamesii* | *An. jamesii* | 6 | 12 | 12 |
| Cellia | Jamesii | *An. splendidus* | *An. splendidus* | 6 | 7 | 8 |
| Cellia | Kochi | *An. kochi* | *An. kochi* | 11 | 20 | 20 |
| Cellia | Leucosphyrus | *An. baimaii* | *An. dirus s.l.* | 6 | 11 | 11 |
| Cellia | Leucosphyrus | *An. baimaii/dirus* | *An. dirus s.l.* | 2 | 2 | 2 |
| Cellia | Maculatus | *An. dravidicus* | *An. dravidicus* | 1 | 2 | 2 |
| Cellia | Maculatus | *An. maculatus* | *An. maculatus* | 9 | 12 | 12 |
| Cellia | Maculatus | *An. pseudowillmori* | *An. pseudowillmori* | 7 | 9 | 9 |
| Cellia | Maculatus | *An. sawadwongporni* | *An. sawadwongporni* | 4 | 3 | 4 |
| Cellia | Subpictus | *An. vagus* | *An. vagus* | 5 | 13 | 13 |
| Cellia | Tessellatus | *An. tessellatus s.l.* | *An. tessellatus s.l.* | 1 | 1 | 1 |
| Cellia | Unclassified | *An. karwari* | *An. karwari* | 1 | 1 | 1 |

## Construction of MSI2 reference database and test panel

From the 214 molecularly identified *Anopheles* specimens, 2140 mass spectra were acquired (Fig 2). A total of 94 spectra (4.4%) were excluded from the MSI2 reference database, based on our predefined criteria for high-quality spectra (see method section, construction of MSI2 reference database). Consequently, the final MSI2 reference database contained 2046 spectra from 209 specimens. The test panel included all 214 molecularly identified specimens, considering only the first four spectra replicates per specimen, for a total of 856 spectra.

## Similarity between mass spectra of MSI2 reference database

The heat map grid of MSI2 score values showed a high degree of reproducibility between mass spectra (Fig 3 and S2 Table). As expected, we observed high MSI2 scores between closely related species belonging to the same species group, indicating an important similarity of mass spectra. Examples include comparisons between *An. dissidens*, *An. campestris/wejchoochotei*, and *An. saeungue* (Barbirostris group); *An. jamesii* and *An. splendidus* (Jamesii group); and *An. sawadwongporni*, *An. dravidicus*, and *An. maculatus* (Maculatus group). Some similarity was also observed between species belonging to the three species groups of the Neocellia series (Annularis, Jamesii and Maculatus groups).

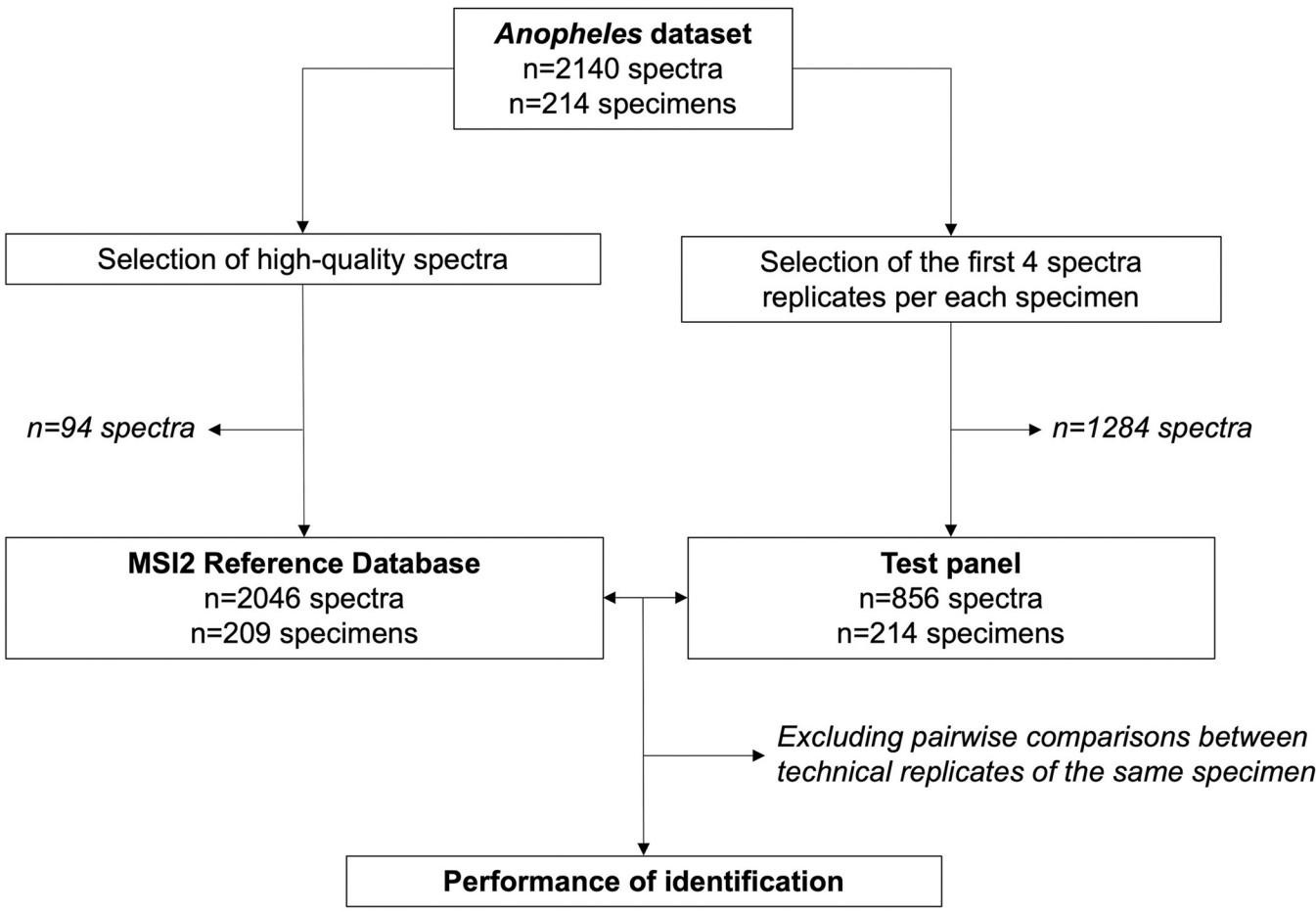

**Fig 2. Study flowchart of the MSI2 reference database construction and validation for MALDI-TOF MS identification of Southeast Asian *Anopheles* species.**

## Performance of MALDI-TOF MS identification

When querying the MSI2 reference database, 831 out of 856 tested spectra provided correct identifications, while 13 spectra yielded incorrect identifications and 12 spectra corresponded to single specimen unreferenced species (Fig 4A and S3 Table). The correct identifications were associated with high scores, shown by a median score (min-max) of 58.12 (13.35–77.55), underscoring the robustness of *Anopheles* mass spectra protein profiles. Lower median scores (min-max) of 28.89 (0–51.62) and 42.60 (31.79–47.73) were obtained for incorrect identifications (t-test, p<0.001) and unreferenced species (t-test, p<0.001), respectively.

When considering only the best result of the four spectra replicates (214 specimens tested), 208 specimens were correctly identified, and only 6 misidentifications were recorded (Fig 4B and S3 Table). The correctly identified specimens predominantly had high identification scores, with a median score (min-max) of 61.62 (15.94–77.55). Additional details for the incorrect answers are provided in Table 2.

When using an identification threshold of 15, the performance of the identification system remained high, with a 97% PPV and 100% of specimens identified (Table 3). When the threshold was increased to 60, the PPV was 100%, but only 59% of the samples could be identified. The optimal balance between the percentage of identified specimens and the reliability of subsequent identification was achieved with a score threshold of 45 (Fig 5). Therefore, we defined

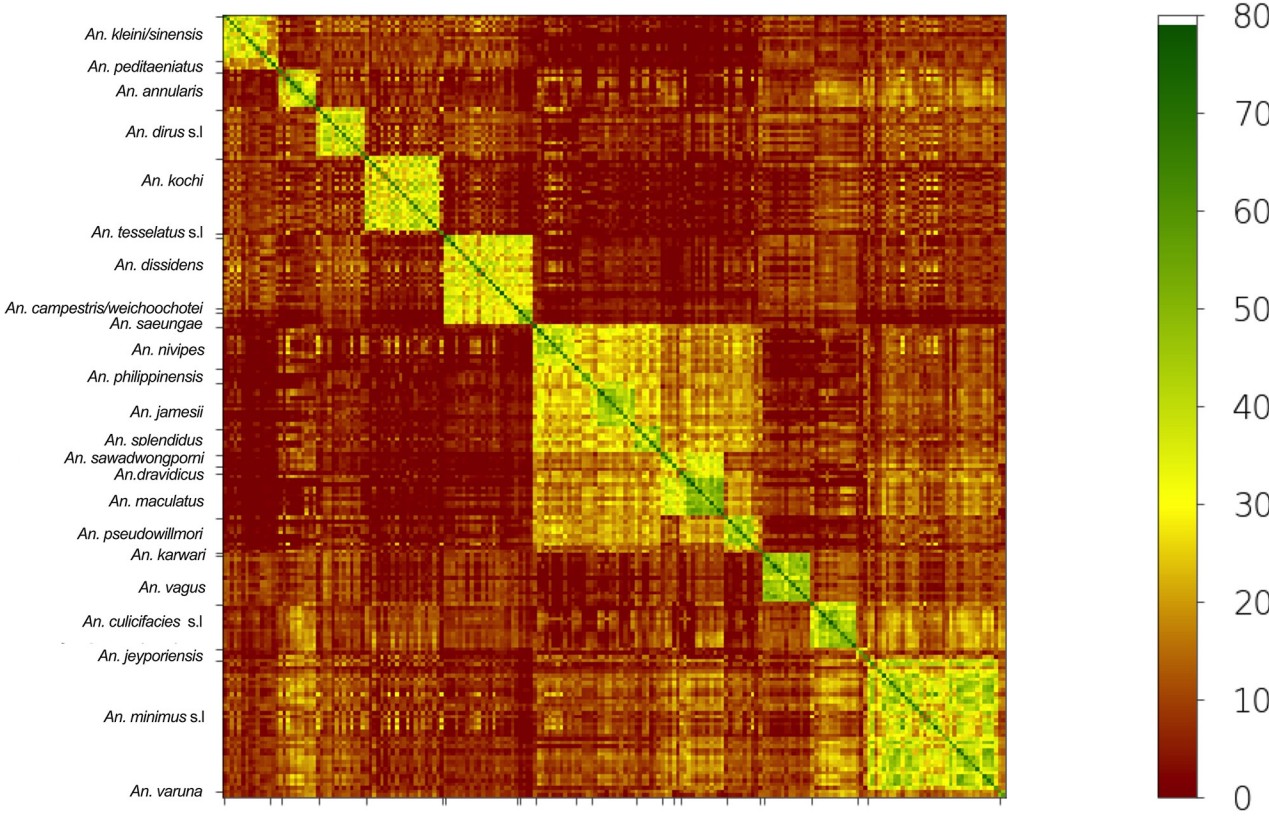

**Fig 3. Heat map grid of the mean score values for specimens in the MSI2 reference database of Southeast Asian *Anopheles* mosquitoes based on the ITS2 classification order.** Similarity levels between mass spectra are colored from green to brown based on the MSI2 score, indicating relatedness and incongruence between spectra, respectively. The green-colored squares along the central diagonal reflect the high degree of reproducibility between mass spectra replicates of the same specimen (intra-specimen reproducibility). Adjacent to the central diagonal, yellow-colored squares indicate a strong level of reproducibility between spectra from different specimens of the same species (inter-specimen reproducibility). Outside the diagonal area, brown colors indicate low similarity scores between the spectra of different species, highlighting the high intra-species specificity of the mass spectra.

the threshold for an interpretable identification result as the best identification score of four replicates with a score > = 45.

Using this threshold of 45, 201/214 (93.9%) of the samples in the test panel had an identification score above the threshold. Of these, 98.5% of the samples were correctly identified (198/

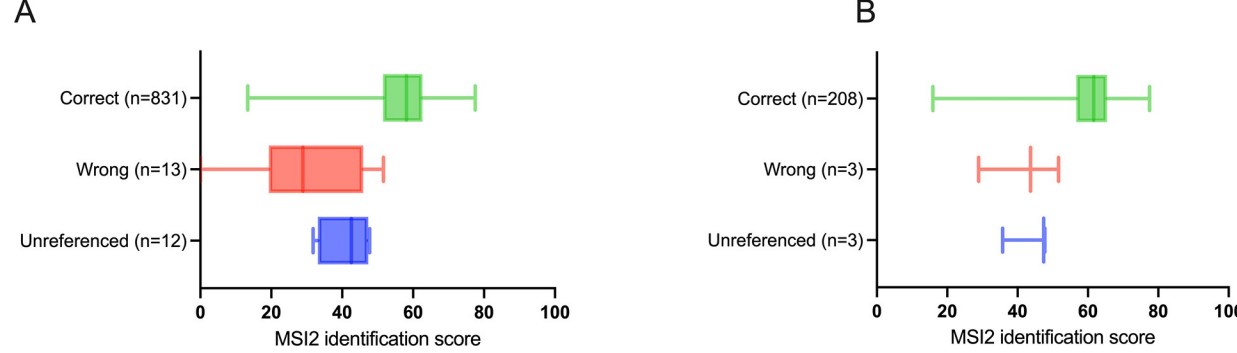

**Fig 4. Distribution of MALDI-TOF MS identification scores when querying of the MSI2 reference database.** (A) Results of all the tested spectra (n = 856 spectra). (B) Best identification score results of the four replicates of each *Anopheles* specimen tested (n = 214).

**Table 2. MALDI-TOF MS incorrect answers using the MSI2 database for Southeast Asian *Anopheles* mosquitoes, using the best identification score (n = 214).**

| Specimen number | Species (ITS2 and or COI molecular identification) | MALDI-TOF MS result | Best MSI score | MSI2 Reference Database Specimen Number | MALDI-TOF MS interpretation |
|---|---|---|---|---|---|
| METF2_0009003 | *An. tessellatus s.l.* | *An. minimus s.l.* | 47.43 | METF2_0009082 | Species not referenced |
| METF2_0009114 | *An. dissidens* | *An. tessellatus s.l.* | 28.89 | METF2_0009003 | Wrong identification |
| METF2_0013841 | *An. karwari* | *An. pseudowillmori* | 35.77 | METF2_0009022 | Species not referenced |
| METF2_0013929 | *An. jeyporiensis* | *An. annularis s.l.* | 43.69 | METF2_0031588 | Wrong identification |
| METF2_0031571 | *An. dravidicus* | *An. maculatus* | 51.62 | METF2_0031616 | Wrong identification |
| METF2_0031708 | *An. campestris/ wejchoochotei* | *An. dissidens* | 47.73 | METF2_0031716 | Species not referenced |

201). Only three misidentifications were recorded. One specimen of *An. dravidicus* was misidentified as *An. maculatus*, a closely related species of the Maculatus group. Other errors were attributed to rare unreferenced species (*An. campestris/wejchoochotei* and *An. tessellatus* s.l.). Among the 13 unidentified specimens (i.e., with a best score <45), 4 belonged to rare species with 4 or fewer specimens in the MSI2 reference database (*An. jeyporiensis* and *An. sawadwongporni*), 1 belonged to an unreferenced species (*An. karwari*.), and the remaining 8 were well-represented species. For this latter group, the scores below the threshold of 4 specimens (between 20.4 and 34.25) may be attributed to lower quality spectra as these specimens were excluded from the reference database based on our quality selection criteria. The other 4 specimens had scores very close to the threshold (between 39.23 and 44.75). Notably, rare species such as *An. varuna* (2 specimens) and *An. saeungae* (4 specimens) were successfully identified. Only 13/31 *An. minimus* specimens were correctly assigned to clade I (1 specimen) or clade II (11 specimens). Within the Annularis complex, 6/10 specimens were correctly assigned to *An. annularis* s.l., and the remaining specimens were misidentified as either *An. annularis* s.l. or *An.* sp. near *annularis.*

## Discussion

We have successfully developed a MALDI-TOF MS reference database for the identification of *Anopheles* species from Southeast Asia. The database contains 2046 spectra from 209 field mosquito specimens molecularly identified using stringent criteria. Using a test panel (214 specimens) and a score threshold of 45, the database achieved 98.5% PPV (198/201) in identifying *Anopheles* mosquitoes at the species or complex level. A key aspect of this study is the sharing of the reference database online, which is freely accessible after registration on the MSI-2 identification platform [49] following the example of molecular databases.

The high MSI2 scores between specimens of the same species highlight the quality and reproducibility of the mass spectra. This also underscores the intra-species specificity of the mass spectra protein profiles. We specifically utilized mosquito heads and demonstrated their

**Table 3. Performance of the identification system at different MSI2 score thresholds, from 15 to 60, using the best identification score (n = 214).**

| Threshold (> =) | 15 | 20 | 25 | 30 | 35 | 40 | 45 | 50 | 55 | 60 |
|---|---|---|---|---|---|---|---|---|---|---|
| Identification rate | 1 | 0.995 | 0.991 | 0.986 | 0.972 | 0.958 | 0.939 | 0.883 | 0.799 | 0.593 |
| Positive predictive value | 0.972 | 0.972 | 0.972 | 0.976 | 0.976 | 0.980 | 0.985 | 0.995 | 1.00 | 1.00 |
| Number of correct identifications | 208 | 207 | 206 | 206 | 203 | 201 | 198 | 183 | 171 | 127 |
| Number of wrong identifications | 6 | 6 | 6 | 5 | 5 | 4 | 3 | 1 | 0 | 0 |
| Number of specimens not identified | 0 | 1 | 2 | 3 | 6 | 9 | 13 | 30 | 43 | 87 |

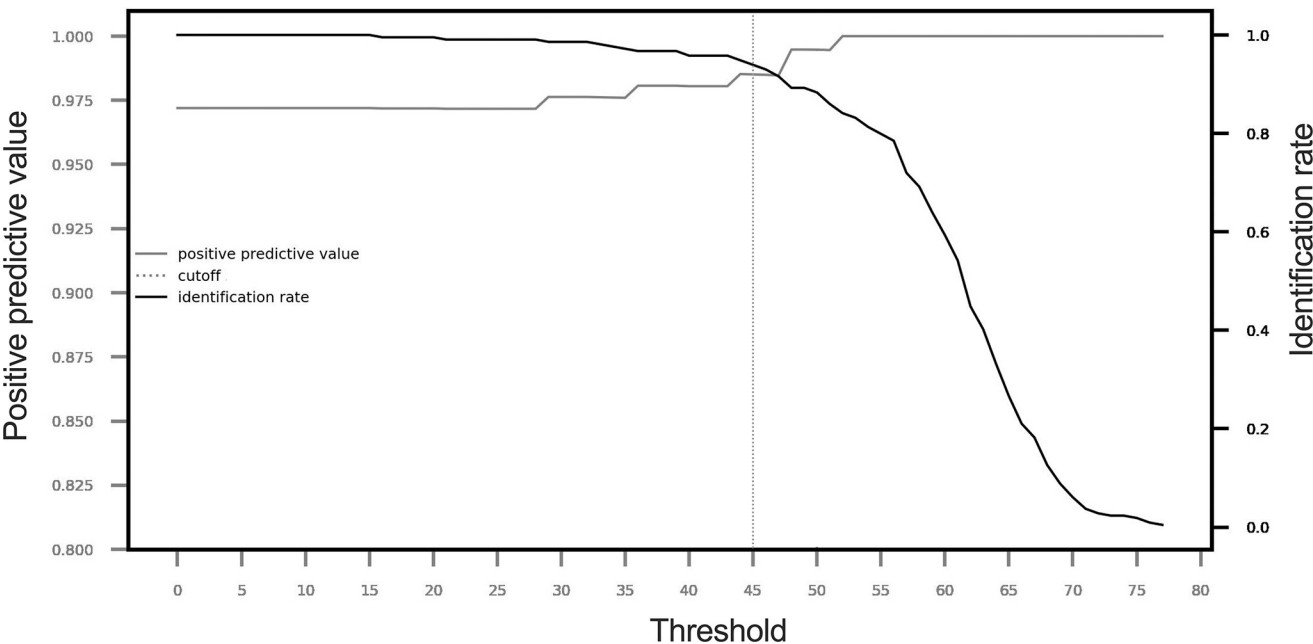

**Fig 5. Determination of the optimal MSI2 score threshold.** Evolution of the identification rate and positive predictive value as a function of the threshold. The analysis was performed considering the best identification score of the four replicates of each *Anopheles* specimen tested (n = 214).

suitability for the identification of field-collected *Anopheles* mosquitoes. Our previous findings indicated higher identification scores using the head in comparison to the thorax and legs of field-collected *Anopheles* mosquitoes [29]. However, conflicting results exist in the literature regarding the most appropriate anatomical part, influenced by laboratory protocols and specimen origins [51]. It is noteworthy that legs, especially when exposed to pyrethroid insecticides, are prone to loss. Pooled specimens during collection, transport, or storage can also impact MALDI-TOF MS identification performance when relying on legs [29]. Furthermore, the thorax may be susceptible to contamination from the blood in the abdomen of engorged specimens during dissection, affecting mass spectra quality [23,29].

Our findings reveal a good concordance between MALDI-TOF MS identification and molecular taxonomic assignment, as determined using ITS2 and COI markers. The heat map grid of mean score values in the MSI2 reference database showed higher similarity scores between spectra of the closest species and species groups on the phylogenetic tree, following the ITS2 classification order. However, our data indicate that the resolution of MALDI-TOF MS, using the MSI-2 algorithm, falls short of DNA sequencing approaches [20]. Specifically, MALDI-TOF MS failed to discriminate spectra at the infra-specific level, as exemplified by the two *An. minimus* lineages. Similar results were observed in studies using the Bruker algorithm for closely related *Anopheles* species in the Gambiae complex [23,29,30]. To improve discrimination among highly similar mass spectra, the implementation of advanced algorithms based on machine learning shows promise. Such algorithms have proven effective in identifying distinctive patterns within mass spectra, offering potential solutions to the challenges associated with discriminating very closely related taxonomic levels [23–25].

In addition to being accessible online on the MSI-2 platform, our database offers the advantage of including a large diversity of Southeast Asian *Anopheles* species. They belong to 20 *sensu stricto* species and 5 sibling species pairs or complexes, which may cover the most common species in this area [10]. Two previous studies have established in-house reference databases for the identification of *Anopheles* species in Southeast Asia using MALDI-TOF MS

[32,33]. Mewara *et al*. focused on mass spectra from *Anopheles* cephalothoraxes field-collected in India, including only two sibling species complexes and one *sensu stricto* species. Huynh *et al*. conducted mass spectra analysis on the legs of *Anopheles* field-collected in Vietnam, covering seven sibling species complexes and six *sensu stricto* species. In our study, using a large test panel (n = 214), we affirm the accuracy of MALDI-TOF MS for *Anopheles* species identification in Southeast Asia.

Notably, among the tested species referenced in the database, we observed only one identification error, between *An. dravidicus* and *An. maculatus*. This discrepancy is probably due to the similarity of spectra between the two specimens, as they belong to the same group and are closely related in the phylogenetic tree. In addition, *An. dravidicus* is a rare species represented by only two specimens in the database, which is insufficient. We have shown that rare species are less accurately identified, probably due to a lack of spectral diversity in the database. The overall high performance of our database can be attributed to stringent molecular identification criteria, ensuring accurate identification of reference species. Additionally, optimal storage conditions of the samples at -80˚C before analysis ensured effective protein preservation. Finally, we developed an innovative decision algorithm to exclude low-quality spectra that might represent a valuable tool for future MALDI-TOF MS studies utilizing the MSI2 software.

This study has several limitations. Because our database was non-exhaustive, it precluded a comprehensive evaluation of performance in discriminating Southeast Asian *Anopheles* sibling species. Many sibling species were absent from the dataset, and certain specimens were molecularly identified only at the complex level. Consequently, we did not assess the accuracy of discriminating sibling species except for some species of the Barbirostris complex. This potential gap could artificially elevate the performance metrics and might impact reproducibility when sample composition varies. In addition, specimens were exclusively collected using the animal-baited trap collection method, resulting in an under-representation of the most anthropophagous *Anopheles* species in our dataset.

In the future, continuous efforts to enrich our MALDI-TOF MS reference database will improve its performance and broaden it applicability across diverse fields. More genera and species of mosquitoes from Southeast Asia should be implemented, using varied trapping methods in different ecological niches and seasons, to introduce greater spectral diversity. The free online availability of databases plays a crucial role, as it allows a larger number of teams to participate in the development and optimization process. This collaborative approach can improve identification accuracy, especially for closely related *Anopheles* species within species complexes. While acknowledging the considerable investment and annual maintenance costs associated with MALDI-TOF MS instruments, it is worth noting that the technique requires minimal reagents, consumables, and technical expertise. In comparison to DNA sequencing, MALDI-TOF MS stands out as a rapid method that allows the identification of *Anopheles* species within few minutes at a cost of 1–2$ per specimen. The analysis of spectra is fully automated and in real-time upon uploading to the MSI-2 identification platform. In addition, MALDI-TOF MS allows a broad range of applications, including the identification of bacteria, fungi, parasites and arthropods, which extends its interest. Given its widespread use in clinical microbiology laboratories, there is a potential for increased availability and adoption for entomological studies.

## Conclusion

MALDI-TOF MS is a valuable tool for the identification of malaria mosquito species, providing a scalable solution for entomological surveillance. This method is particularly relevant in

the context of global change, which requires increased surveillance of mosquito vectors on a large scale. The reference database established in this study is now available to the scientific community through the MSI-2 free online application and will facilitate entomological surveillance of *Anopheles* vector species in Southeast Asia. Continuous efforts to standardize procedures and promote the sharing of databases are essential to expand the use of this tool across different settings. This, in turn, will contribute to the wider adoption and effectiveness of MALDI-TOF MS in entomological surveillance efforts.

## Supporting information

**S1 Fig. Bayesian consensus tree for ITS2 sequences.** The tree is rooted on *Culex pipiens*. Branch are labeled with Bayesian posterior probabilities. The bar represents 0.2 substitutions per site.
(PDF)

**S2 Fig. Bayesian consensus tree for COI sequences.** The tree is rooted on *Culex pipiens*. Branch are labeled with Bayesian posterior probabilities. The bar represents 0.02 substitutions per site.
(PDF)

**S1 Table. List of *Anopheles* specimens included in this study and taxonomic assignment.**
(XLSX)

**S2 Table. Raw data of the heat map grid mean score values for the specimens in the MSI2 reference database, ITS2 classification order.**
(XLSX)

**S3 Table. Raw data of the MALDI-TOF MS identification results after the query of the test panel against MSI2 reference database.**
(XLSX)

## Acknowledgments

We thank the teams of the Entomology and Malaria departments of the Shoklo Malaria Research Unit for their work. The Shoklo Malaria Research Unit is part of the Mahidol-Oxford Research Unit supported by Wellcome (U.K.). A CC BY or equivalent licence is applied to the author accepted manuscript arising from this submission, in accordance with the grant's open access conditions.

## Author Contributions

**Conceptualization:** Victor Chaumeau, Martine Piarroux, François Nosten, Cécile Nabet.

**Data curation:** Victor Chaumeau, Martine Piarroux, Cécile Nabet.

**Formal analysis:** Victor Chaumeau, Martine Piarroux, Cécile Nabet.

**Funding acquisition:** Victor Chaumeau, François Nosten.

**Investigation:** Victor Chaumeau, Martine Piarroux, Thithiworada Kulabkeeree, Sunisa Sawasdichai, Aritsara Inta, Cécile Nabet.

**Methodology:** Victor Chaumeau, Martine Piarroux, Renaud Piarroux, Cécile Nabet.

**Project administration:** Victor Chaumeau, Cécile Nabet.

**Resources:** Wanitda Watthanaworawit, François Nosten, Renaud Piarroux.

**Supervision:** Victor Chaumeau, Cécile Nabet.

**Validation:** Victor Chaumeau, Martine Piarroux, Renaud Piarroux, Cécile Nabet.

**Visualization:** Victor Chaumeau, Martine Piarroux, Renaud Piarroux, Cécile Nabet.

**Writing – original draft:** Victor Chaumeau, Cécile Nabet.

**Writing – review & editing:** Victor Chaumeau, Martine Piarroux, Thithiworada Kulabkeeree, Sunisa Sawasdichai, Aritsara Inta, Wanitda Watthanaworawit, François Nosten, Renaud Piarroux, Cécile Nabet.

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
