## [Decision Letter · Decision Letter 0]

16 Apr 2024

PONE-D-24-11167Identification of Southeast Asian Anopheles mosquito species with MALDI-TOF mass spectrometryPLOS ONE

Dear Dr. Chaumeau,

Thank you for submitting your manuscript to PLOS ONE. After careful consideration, we feel that it has merit but does not fully meet PLOS ONE’s publication criteria as it currently stands. Therefore, we invite you to submit a revised version of the manuscript that addresses the points raised during the review process.

We look forward to receiving your revised manuscript.

Kind regards,

Joseph Banoub, Ph,D., D. Sc.

Academic Editor

PLOS ONE

Journal Requirements:

 “This research was partly funded by Wellcome (#220211).”

“This research was partly funded by Wellcome  (#220211).”

“This research was partly funded by Wellcome  (#220211).”

Additional Editor Comments (if provided):

Please correct according to the minor suggestions recommended.

Reviewers' comments:

Reviewer's Responses to Questions

**Comments to the Author**

1. Is the manuscript technically sound, and do the data support the conclusions?

Reviewer #1: Yes

Reviewer #2: Yes

2. Has the statistical analysis been performed appropriately and rigorously? 

Reviewer #1: Yes

Reviewer #2: Yes

3. Have the authors made all data underlying the findings in their manuscript fully available?

Reviewer #1: Yes

Reviewer #2: Yes

4. Is the manuscript presented in an intelligible fashion and written in standard English?

Reviewer #1: Yes

Reviewer #2: Yes

5. Review Comments to the Author

Reviewer #1: An interesting piece of work. It produces a widely available database that should prove of considerable value in analysis and control of mosquito thence malaria problems. There are some minor quibbles, e.g. does political instability in Myanmar cause increase in P.falsiparum and P.vivax due to changes in human migration, mosquito movement or change in control measures?, also it is implied that not only does MALDI-Tof analysis identify mosquito species but also infection states, past blood meal and mosquito age. However the overall technical description is clear and the data obtained justify the claims made. The realism expressed in the discussion as to how the identification procedure can be refined in the future is welcome; the comparison with previous efforts using smaller sample sizes and species types, plus the use of head extracts as opposed to other body parts and the benefits therein validate the value of this new data. The need for optimal sample storage and multiple specimens is correctly emphasized. The development of the algorithm and use to reject lower quality spectra is an integral part of the data appreciation, at first consideration the selection of 45 as the score threshold seemed counter-intuitive but detailed consideration of the data provided demonstrated the correctness of the selection.

Reviewer #2: Dear author,

Please find below some comments on your manuscript:

p. 14, line 116: please mention the MALDI matrix used and the amount of protein extract spotted in the target plate.

p. 16, line 152: more details are needed, like the noise filtering, intensity threshold, de-isotoping…

p. 21, line 220: please precise the “predefined criteria for high quality spectra”

Best,

6. PLOS authors have the option to publish the peer review history of their article (what does this mean?). If published, this will include your full peer review and any attached files.

Reviewer #1: No

Reviewer #2: No

---

## [Author Response · Author response to Decision Letter 0]

20 May 2024

Dear Editor,

We are thankful for the review of the manuscript “Identification of Southeast Asian Anopheles mosquito species using MALDI-TOF mass spectrometry”. Below is a point-by-point answer to Editor’s and Reviewers’ comments:

Answer: The manuscript was revised to meet PLOS ONE’s style requirements.

2. Please provide an amended statement that declares *all* the funding or sources of support (whether external or internal to your organization) received during this study. Please also include the statement “There was no additional external funding received for this study.” in your updated Funding Statement.

Answer: The funding statement was amended to meet the journal requirements. The amended funding statement should read:

“This research was funded by Wellcome (#220211), the Bill and Melinda Gates Foundation (#OPP1177406) and the Global Fund (#QSE-M-UNOPS-20864-003-32). There was no additional external funding received for this study.”

3. We note that you have provided additional information within the Acknowledgements Section that is not currently declared in your Funding Statement. Please remove any funding-related text from the manuscript and let us know how you would like to update your Funding Statement.

Answer: Funding-related text was removed from the manuscript. The amended funding statement is provided above.

4. We note that Figure 1 in your submission contain [map/satellite] images which may be copyrighted. We require you to either (1) present written permission from the copyright holder to publish these figures specifically under the CC BY 4.0 license, or (2) remove the figures from your submission.

Answer: Base map of Figure 1 come from Openstreetmap, which is free to use, according to journal requirements. As requested by the journal, we have written the sentence “Base map and data from OpenStreetMap and OpenStreetMap Foundation”in Figure 1 legend. 

Answer: The reference list was reviewed to meet the journal requirements. Additional references were cited in the Introduction section to give more background information on malaria on the Thailand-Myanmar border (references 3-12 in the revised version of the manuscript).

Reviewers’ comments:

Reviewer #1:

An interesting piece of work. It produces a widely available database that should prove of considerable value in analysis and control of mosquito thence malaria problems. There are some minor quibbles, e.g. does political instability in Myanmar cause increase in P.falsiparum and P.vivax due to changes in human migration, mosquito movement or change in control measures?, also it is implied that not only does MALDI-Tof analysis identify mosquito species but also infection states, past blood meal and mosquito age. However the overall technical description is clear and the data obtained justify the claims made. The realism expressed in the discussion as to how the identification procedure can be refined in the future is welcome; the comparison with previous efforts using smaller sample sizes and species types, plus the use of head extracts as opposed to other body parts and the benefits therein validate the value of this new data. The need for optimal sample storage and multiple specimens is correctly emphasized. The development of the algorithm and use to reject lower quality spectra is an integral part of the data appreciation, at first consideration the selection of 45 as the score threshold seemed counter-intuitive but detailed consideration of the data provided demonstrated the correctness of the selection.

Answer to reviewer #1:

We thank the reviewer for her/his constructive feed-back on the manuscript. The manuscript was revised to address the points raised by the reviewer:

“There are some minor quibbles, e.g. does political instability in Myanmar cause increase in P.falsiparum and P.vivax due to changes in human migration, mosquito movement or change in control measures?”

Answer: Additional details were provided in the Introduction section to clarify our current understanding of how political instability in Myanmar affects malaria epidemiology in Karen state. The surge in malaria observed in recent years in Karen state is multi-factorial and remains largely unexplained. Entomological factors may be involved but there is currently no data available to support this hypothesis. Entomological factors are unlikely directly affected by the political context. Political instability has led to disruption of health services across the country and affected human behaviours. More specifically, the armed-conflict has disrupted access to early diagnosis and treatment (the main pillar of falciparum malaria elimination in this setting) and likely increased human exposures to vector bites (because fleeing civilians sought temporary shelters in forested areas and on the river banks which delimit the international border between Thailand and Myanmar, two typical habitats of the main local vectors) in the fighting areas. Mosquito bed nets have only a marginal impact on malaria in this region because the vectors bite mostly outdoors and at a time when people are not protected by mosquito bed nets; the efficacy of mosquito bed nets for malaria vector control is unlikely affected by political instability in Myanmar. Parasitological factors including a shift in Plasmodium spp. resistance to antimalarial drugs may also be involved and are the focus of active research.

“Also it is implied that not only does MALDI-Tof analysis identify mosquito species but also infection states, past blood meal and mosquito age.”

Answer: Assessment of infection states, past blood meal and mosquito age typically involves processing of different anatomical parts (e.g. the thorax, legs and abdomen) and analysing the data with protein profiling and machine learning. Additional details can be found in the cited references. These potential applications are out of the scope of the current study (mosquito species identification); therefore, they were not discussed in more details as part of this article.

Reviewer #2:

Dear author,

Please find below some comments on your manuscript:

p. 14, line 116: please mention the MALDI matrix used and the amount of protein extract spotted in the target plate.

p. 16, line 152: more details are needed, like the noise filtering, intensity threshold, de-isotoping…

p. 21, line 220: please precise the “predefined criteria for high quality spectra”

Answer to reviewer #2:

We thank the reviewer for her/his constructive feed-back on the manuscript. The manuscript was revised to address the points raised by the reviewer:

“p. 14, line 116: please mention the MALDI matrix used and the amount of protein extract spotted in the target plate.”

Answer: We have not performed a quantification of the proteins in the protein extracts to be spotted because this information is not useful to ensure quality mass spectra using MALDI-TOF. Indeed, we ensured a reproducible quality of mass spectra using our protocol without any quantification of proteins. 

“p. 16, line 152: more details are needed, like the noise filtering, intensity threshold, de-isotoping…”

Answer: The pre-processing of mass spectra is standard, it includes a smoothing (python library pimzml), a baseline correction and a custom peak picking with selection of the 70 highest intensity peaks. Noise filtering and intensity threshold are not applied here, because there is a ranking of the highest intensity peaks instead. The de-isotoping is not applied using MALDI-TOF mass spectrometry. A specific comment has been added at lines 181-183 (version with tracked changes) of the method section. 

“p. 21, line 220: please precise the “predefined criteria for high quality spectra”

Answer: The predefined criteria for high quality spectra are mentioned in the method section, in the paragraph about the construction of MSI2 reference database 189-197 (version with tracked changes). We referred to this method section after the cited sentence at lines 255 (version with tracked changes).

Sincerely,

Dr. Victor Chaumeau

---

## [Editor Report · Decision Letter 1]

27 May 2024

Identification of Southeast Asian Anopheles mosquito species with MALDI-TOF mass spectrometry

PONE-D-24-11167R1

Dear Dr. Chaumeau,

We’re pleased to inform you that your manuscript has been judged scientifically suitable for publication and will be formally accepted for publication once it meets all outstanding technical requirements.

Kind regards,

Joseph Banoub, Ph,D., D. Sc.

Academic Editor

PLOS ONE

Additional Editor Comments (optional):

Thank you very much for completing all the revisions required.
---

## [Editor Report · Acceptance letter]

31 May 2024

PONE-D-24-11167R1 

PLOS ONE

Dear Dr. Chaumeau, 

I'm pleased to inform you that your manuscript has been deemed suitable for publication in PLOS ONE. Congratulations! Your manuscript is now being handed over to our production team.

Kind regards, 

on behalf of

Dr. Joseph Banoub 

Academic Editor

PLOS ONE